# The Unique Cauda-Liked Structure Represents a New Subfamily of Cunaxidae: Description of New Taxa and Discussion on Functional Morphology [note 1]

**DOI:** 10.3390/ani13081363

**Published:** 2023-04-15

**Authors:** Jianxin Chen, Maoyuan Yao, Jianjun Guo, Tianci Yi, Daochao Jin

**Affiliations:** 1Institute of Entomology, Guizhou University, Guiyang 550025, China; jianxinchen000@163.com (J.C.);; 2The Guizhou Provincial Key Laboratory for Plant Pest Management of Mountainous Region, Guiyang 550025, China; 3The Scientific Observing and Experimental Station of Crop Pest in Guiyang, Ministry of Agriculture, Guiyang 550025, China; 4College of Agriculture, Anshun University, Anshun 561000, China

**Keywords:** Acariformes, Bdelloidea, taxonomy, predator, China

## Abstract

**Simple Summary:**

In this study, a cauda-like structure found in Cunaxidae is defined, and with it the new taxa, Cunaxicaudinae Chen & Jin subfam. nov., and its two new genera, *Cunaxicaudus* Chen & Jin gen. nov. (type genus) and *Brevicaudus* Chen & Jin gen. nov., are erected. It is proposed that the specialized cauda may be the result of the evolution of the sperm transfer mode.

**Abstract:**

A cauda-like structure was found, firstly in Cunaxidae, and with it the new taxa Cunaxicaudinae Chen & Jin subfam. nov., and its two new genera, *Cunaxicaudus* Chen & Jin gen. nov. (type genus) and *Brevicaudus* Chen & Jin gen. nov., were erected. Cunaxicaudinae Chen & Jin subfam. nov. differs from the known members of the family Cunaxidae by the unique conspicuous cauda derived from the posterior end of the hysterosoma. The generic features of *Cunaxicaudus* Chen & Jin gen. nov. are as follows: the posterior of the hysterosoma elongated as a much longer cauda; palp between genu and tibiotarsus without apophysis; *e1* closer to *d1* than *f1*; and *e1* closer to mid-line than *c1* and *d1*. The generic features of *Brevicaudus* Chen & Jin gen. nov. are as follows: the posterior of hysterosoma elongated as a short cauda; palp between genu and tibiotarsus with one apophysis; distance between setae *e1* and *d1* approximately equal to *e1*; and *f1*, *e1* as close to mid-line as *c1* and *d1* to mid-line. It is proposed that the specialized cauda may be the result of the evolution of the sperm transfer mode.

## 1. Introduction

Cunaxidae (Prostigmata: Bdelloidea) erected by Thor [1] is a predatory mite group that can prey on phytophagous mites, other small arthropods and nematodes, etc. [2,3,4,5,6,7]. They commonly inhabit various terrestrial habitats, including forest leaf litter and soil, tree holes, moss, etc., and being important predators, they play a crucial role in agricultural ecosystems [7,8,9]. 

According to the recent literature, there are six subfamilies, 30 genera and more than 450 species described as Cunaxidae in the world [10,11,12,13,14,15,16,17,18,19,20,21]. In this work, with unique hysterosoma, a new subfamily with two new genera and three new species is described in Mohan Port in Mengla county, Xishuangbanna Dai Autonomous Prefecture, Yunnan province, China. Skvarla et al. [10] reviewed the Cunaxidae with keys to the world subfamilies, genera and species. Here, we provide an updated key to the subfamilies of the Cunaxidae to include the new subfamily.

The following abbreviations are used: prodorsum: anterior trichobothria (*at*), posterior trichobothria (*pt*), lateral proterosomal (*lps*), median proterosomal (*mps*); hysterosoma: internal humerals (*c1*), external humerals (*c2*), internal dorsals (*d1*), internal lumbals (*e1*), internal sacrals (*f1*), external sacrals (*f2*), internal clunals (*h1*), external clunals (*h2*); venter: propodogastral seta (*ppgs*), hysterogastral seta (*hgs*); anal region: pseudanal (*ps*); genital region: aggenitals (*ag*), genitals (*g1–4*); gnathosoma: hypognathals (*hg1–4*); leg: attenuate (sharply) solenidion (*asl*), blunt-pointed rod-like solenidion (*bsl*), famulus (*fam*), trichobothria (*T*), simple tactile seta (*sts*), microseta (*mst*), dorsoterminal solenidion (*dtsl*). Duplex and triplex setae are indicated in brackets ({}).

## 2. Materials and Methods

### 2.1. Sampling Area

Samples of fallen leaves were collected from the woodland at Mohan Port (21°11′22.66″ N, 101°41′51.80″ E, elevation 893 m) in Mengla county, Xishuangbanna Dai Autonomous Prefecture, Yunnan province, China.

Mohan Port, bordering the Botan port of Laos, is the only national port between China and Laos. Mohan Port, bordering the Botan port of Laos, is the only national port between China and Laos, where the climate is pleasant: there is no chilly winter and hot summer, four seasons is not clear but raining season with the distinction of the dry quarter, and the average annual temperature is 21.2 °C. The rainfall here averages 1615 mm.

### 2.2. Laboratory Activities

Fallen leaves were placed in a modified Berlese-Tullgren funnel for at least eight hours to isolate mite specimens. The specimens were preserved in 75% alcohol and then mounted on slides in Hoyer’s medium [22]. Coordinates and altitudes were obtained by smartphone with GPS. Line drawings were produced with the aid of a drawing tube attached to a phase contrast Nikon Ni E microscope with DIC optics, and photographs were taken using a camera (Nikon DS-Ri 2) attached to a Nikon Ni E microscope with DIC optics. All figures were edited with Adobe Photoshop CC 2019. All measurements were taken with the software Nikon NIS Elements AR 4.50 and provided in µm for the holotype and paratypes in parentheses. The nomenclature and abbreviations of idiosoma follow Den Heyer and Castro [23] and Skvarla et al. [10], except for propodosomal setae, which follows Fisher et al. [24], and legs setal notation follows Den Heyer [25].

## 3. Results

Family Cunaxidae Thor, 1902; Subfamily Cunaxicaudinae Chen & Jin subfam. nov.; Type genus: *Cunaxicaudus* Chen & Jin gen. nov.

The new subfamily was established by diagnostic caudal structure, which consists of three sections: caudal base, caudal petiole and caudal xiphoid (Figure 1, Figure 2, Figure 3B and Figure 4).

Caudal base: the extended posterior end of the hysterosoma, with the genital region, and with no suture separating the hysterosoma from the caudal base, was present but clearly less sclerotized than the main hysterosoma.

Caudal petiole: a tubelike extension from the caudal base, weakly sclerotized, translucent or transparent, with a distinct suture line present between it and the caudal base.

Caudal xiphoid: the extension from the caudal petiole, sword shaped and transparent, with a clear suture line demarcating it from the caudal petiole.

The subfamily Cunaxicaudinae Chen & Jin subfam. nov. can be easily distinguished from other members of the family Cunaxidae by the unique caudal structure. According to the literature, in most cunaxids sperm transfer is indirect, but in some species it is direct by mating with a visible aedeagus. Therefore, we infer that the caudal structure may be more conducive to the cunaxid’s mating.

Etymology. The new subfamily is named from the stem of Cunaxidae (Cunaxi-), and the posterior of hysterosoma being noticeably elongated as a cauda (-caudinae), which means the tail or tail-like structure of an animal, bird, fish, or other creature in Latin.

### 3.1. Cunaxicaudus Chen & Jin gen. nov.

Species Type: *Cunaxicaudus mohanensis* Chen & Jin sp. nov.

Generic features (male): posterior of hysterosoma is much elongated as a conspicuous cauda; the caudal petiole and caudal xiphoid are long; palp between genu and tibiotarsus without apophysis; distance between setae *e1* and *d1* about 1/3 of that between *e1* and *f1*, and *e1* closer to mid-line than *c1* and *d1*; lyrifissures (*im*) close to and at the same level as *f1*; leg IV longest and leg II shortest, and tarsal lobes well-developed.

#### 3.1.1. *Cunaxicaudus mohanensis* Chen & Jin sp. nov.

Diagnosis. The *h1* was longer than other dorsal setae (*lps*, *mps*, *c1*, *c2*, *d1*, *e1*, *f1*); two pairs of hysterogastral setae (*hgs1–hgs2*); basifemora I–IV: 4-4-3-0 *sts*.

Description (Figure 1, Figure 2, Figure 3, Figure 4, Figure 5, Figure 6, Figure 7 and Figure 8); male (*n* = 17).

The idiosoma length was 324 (305–364) from the base of subcapitulum to the posterior edge of median shield, and the width was 188 (170–219); the posterior end of the hysterosoma was elongated as a very long cauda (Figure 1, Figure 2 and Figure 4).

Dorsum (Figure 2 and Figure 3A,B). Propodosomal and hysterosomal shields entirely complemented by reticulations, integument with striae. The propodosomal shield was 101 (101–128) long and 155 (135–160) wide, sclerotized and with a reticulated pattern, and was bearing one pair of anterior (*at*) and one pair of posterior (*pt*) setose trichobothria and two pairs of tactile setae (*lps* and *mps*); *at* was shorter than the length of *pt*, *lps* near *pt* base; the area was anterior to *at* papillary. The lengths of setae and the distances between the bases of setae were *at* 151 (145–170), *pt* 178 (160–220), *lps* 7 (5–7), *mps* 6 (5–6); *at-at* 18 (12–19), *pt-pt* 110 (83–128), *lps-lps* 109 (84–118), *mps-mps* 46 (44–57), *lps-mps* 39 (27–40), *at-lps* 69 (59–74), *pt-mps* 35 (24–38), *pt-lps* 16 (14–24), *at-mps* 77 (67–83) and *at-pt* 86 (74–93).

The hysterosomal median shield was 123 (99–135) long, 134 (110–140) wide, with five pairs of dorsal setae (*c1*, *c2*, *d1*, *e1*, *f1*) and one pair of lyrifissures (*im*) close to and at same level with *f1*. The distance between setae *e1* and *d1* was about one-third of that between *e1* and *f1*; *e1* was closer to the mid-line of the median shield than *c1* and *d1*. Setae *h1* was situated on the striated integument of the caudal base and was longer than other dorsal setae. The lengths of six pairs of dorsal setae were *c1* 6 (5–8), *c2* 5 (5–8), *d1* 7 (5–7), *e1* 7 (5–9), *f1* 8 (8–10), *h1* 15 (11–16). Distances of setae: *c1-c1* 63 (66–77), *c2*-*c2* 117 (113–129), *d1*-*d1* 66 (53–68), *e1*-*e1* 43 (39–45), *f1*-*f1* 40 (37–42), *h1*-*h1* 16 (11–19), *c1*-*c2* 30 (26–36), *c1*-*d1* 39 (30–46), *c2*-*d1* 38 (36–44), *d1*-*e1* 14 (12–18), *e1*-*f1* 37 (33–38) and *f1*-*h1* 30 (26–38).

The *Cauda dorsum* is shown in Figure 1, Figure 2 and Figure 3B. From the dorsal view, the posterior end of the idiodoma was elongated significantly as a long cauda, clearly defined and gradually narrowed, consisting of a caudal base with light striae, a caudal petiole with transverse plicated striae on edge, and a smooth caudal xiphoid. The cauda was 290 (255–303) long, measured from the posterior edge of the median shield to the end; the caudal base was 80 (78–108), the caudal petiole was 125 (80–132) and the caudal xiphoid was 85 (75–90). The anal region was dorsally located on the caudal base with dotted fine papillae, bearing two pairs of pseudanal setae (*ps1*–*ps2*), 4 (8–15) and 10 (8–14) in length, respectively; one pair of *h2* was 10 (8–11) in length and one pair of lyrifissures (*ih*), and the rest of the caudal base was also covered with fine papillae, bearing genital setae *g3*–*g4* representing the genital region.

*Venter* (Figure 3C, Figure 4 and Figure 5). Coxae I–IV fused, resulting in a clearly complete ventral shield with dotted papillae; coxae III–IV with visible reticulated pattern. There was one pair of propodogastral setae (*ppgs*) 6 (5–7) close to coxae II and two pairs of hysterogastral setae (*hgs1*–*hgs2*), and the lengths of setae *hgs1*–*hgs2* were 7 (6–8) and 21 (15–21); one pair of clear foveolae was medially located between the coxae III groups. The area between hysterogastral setae *hgs2* with transverse striation. The setal formula of coxal plates I–IV was 3-1-3-3 *sts*.

*Cauda venter* (Figure 4 and Figure 5). The cauda was 290 (267–283) long: the caudal base was 81 (89–105), including the genital area; the caudal petiole with plicated striae was 124 (82–129) long; the caudal xiphoid was smooth and 85 (79–90) long. The cauda was clearly defined, and gradually narrowed as in its dorsal view. The caudal base had horizontal striation except for the genital region (genital shield); the genital region had dotted papillae, genital suckers (two pairs) visible, and four pairs of genital setae (*g1*–*g4*), of which *g3*–*g4* were dorsally located (Figure 2 and Figure 3B). Lengths of setae *g1*–*g4* were 15 (12–15), 16 (11–18), 20 (12–21) and 17 (13–20), respectively.

*Gnathosoma*. Palp (Figure 6 and Figure 7A). Five-segmented, 136 (117–144) long, with granulated papillae and terminated with a bifid claw. Palp chaetotaxy was as follows: trochanter without setae; basifemur with one dorsal simple seta; telofemur with one dorsal stout seta and one short and tapering apophysis; genu with two stout setae and two simple setae; and tibiotarsus with one stout seta, three simple setae and one acute solenidion.

Chelicerae (Figure 7B): 107 (98–132) long, with fine papillae and a reticulated pattern; one cheliceral seta, 8 (7–10) in length; terminating in a well developed chela.

Subcapitulum (Figure 6 and Figure 7C) had dotted papillae, 128 (109–132) long and 69 (67–84) wide; two pairs of apophyses, of which one pair was smaller and claw-like and another pair was blunt rod-like; two pairs of adoral setae (*ads1*, *ads2*), of which *ads1* 6 (6–9) and *ads2* were 3 (3–5) in length; four pairs of hypostomal setae (*hg1*–*hg4*), where *hg2* and *hg4* were subequal in length and both four times longer than *hg1* and *hg3*; and lengths of *hg1*–*hg4* of 11 (10–13), 36 (33–53), 6 (6–9) and 37 (28–43), respectively. The distances of paired *hg*-setae were *hg1-hg1* 5 (5–7), *hg2-hg2* 14 (12–15), *hg3-hg3* 28 (25–31), *hg4-hg4* 55 (52–65), *hg1-hg2* 10 (9–12), *hg2-hg3* 53 (46–58) and *hg3-hg4* 19 (15–23).

*Legs* (Figure 8). Leg IV was the longest and leg II was the shortest, with tarsal lobes fairly well-developed; the dorsum of all the leg segments had finely granulated papillae. Lengths of legs I–IV: 246 (217–259), 210 (198–224), 280 (238–280), 300 (261–309). Lengths of tarsi I–IV: 93 (80–99), 77 (70–84), 108 (91–113) and 116 (93–122). Leg I–IV chaetotaxy (including coxae) was as follows: coxae I–IV 3-1-3-3 *sts*; trochanters I–IV 1-1-2-1 *sts*; basifemora I–IV 4-4-3-0 *sts*; telofemora I–IV 4-4-4-4 *sts*; genua I–IV 2 *asl*, {1 *asl*, 1 long *bsl*, 1 *mst*}, 4 *sts*-1 *asl*, 1 long *bsl*, 5 *sts*-1 *asl*, 1 long *bsl*, 5 *sts*-1 *asl*, 1 long *bsl*, 5 *sts*; tibiae I–IV 1 long *bsl*, {1 *asl*, 1 *mst*}, 4 *sts*-1 *bsl*, 5 *sts*-1 *bsl*, 5 *sts*-1 *T*, 4 *sts*; and tarsi I–IV 3 *asl*, 1 long *bsl*, 1 *fam*, 1 *dtsl*, 17 *sts*-1 long *bsl*, 1 *dtsl*, 17 *sts*-1 *dtsl*, 15 *sts*-1 *dtsl* and 14 *sts*. 

Female and other developmental stages: unknown.

Remarks: The new species is distinguished from other known species by its unique cauda-like structure.

Material examined: Holotype, male, collected from fallen leaves, Mohan Port (21°11′22.66″ N, 101°41′51.80″ E, elevation 893 m), Mengla County, Xishuangbanna Dai Autonomous Prefecture, Yunnan Province, China, on 20 November 2018, collector, Jianxin Chen and Xuesong Zhang, slide No. YN-CU-201811201001. Three paratype males, with the same data as the holotype, slides No. YN-CU-201811201002, YN-CU-201811202001 and YN-CU-201811201301. The paratypes, six males, were collected from fallen leaves in Mohan Port (21°11′22.66″ N, 101°41′51.80″ E, elevation 893 m), Mengla County, Xishuangbanna Dai Autonomous Prefecture, Yunnan Province, China, on 6 June, 2019, collector, Jian-Xin Chen, slides No. YN-CU-201906060301–YN-CU-201906060306. The paratype, one male, was collected from fallen leaves in Mohan Port (21°11′22.66″ N, 101°41′51.80″ E, elevation 893 m), Mengla County, Xishuangbanna Dai Autonomous Prefecture, Yunnan Province, China, on 7 June 2019, collector, Jianxin Chen, slide No. YN-CU-201906070201. Seven paratypes, male, collected from fallen leaves in Mohan Port (21°11′22.66″ N, 101°41′51.80″ E, elevation 893 m), Mengla County, Xishuangbanna Dai Autonomous Prefecture, Yunnan Province, China, on 9 June 2019 by Jian-Xin Chen, slide No. YN-CU-201906090201–YN-CU-201906090207. All types of materials were deposited at the Institute of Entomology, Guizhou University, Guiyang, P. R. China (GUGC).

Etymology. The new genus name is derived from the subfamily name Cunaxicaudinae subfam. nov., as above; the new species name refers to Mohan Port where the types were originated.

#### 3.1.2. *Cunaxicaudus neomohanensis* Chen & Jin sp. nov.

Diagnosis. Setae *h1* were longer than other dorsal setae (*lps*, *mps*, *c1*, *c2*, *d1*, *e1*, *f1*) in length, with two pairs of hysterogastral setae (*hgs1*–*hgs2*); basifemora I–IV: 4-4-3-0 *sts*. The cauda was short, and the caudal petiole was wider and the caudal xiphoid was shorter (compared to *C. mohanensis* Chen & Jin sp. nov.). 

Description (Figure 9, Figure 10, Figure 11, Figure 12 and Figure 13); male (*n* = 1).

The idiosoma length was 276 from the base of the subcapitulum to the posterior edge of the median shield and the width was 189; the posterior end of the hysterosoma was elongated, forming a long cauda (Figure 9 and Figure 10). 

*Dorsum* (Figure 9). The propodosomal and hysterosomal shields were entirely complemented by reticulations, surrounding integument striate. The propodosomal shield was 81 long, 130 wide, sclerotized and with a reticulated pattern, and had one pair of anterior (*at*) and one pair of posterior (*pt*) setose trichobothria and two pairs of tactile setae (*lps* and *mps*); *at* was shorter than *pt*, and *lps* was near the *pt* base; the area anterior to *at* was covered with papillae. Lengths of the setae and distances between the bases of setae were *at* 150, *pt* 182, *lps* 7, *mps* 6; *at-at* 23, *pt-pt* 103, *lps-lps* 98, *mps-mps* 45, *lps-mps* 34, *at-lps* 56, *pt-mps* 28, *pt-lps* 22, *at-mps* 65 and *at-pt* 79. 

The hysterosomal median shield had five pairs of simple setae (*c1*, *c2*, *d1*, *e1*, *f1*) and one pair of lyrifissures (*im*) closer to and at same level as *f1*. The distance between setae *e1* and *d1* was about one-third of that between *e1* and *f1*; *e1* was closer to the mid-line of the median shield than *c1* and *d1*. Setae *h1* was situated on the striated integument of the caudal base and was longer than other dorsal setae. The lengths of six pairs of dorsal setae were *c1* 8, *c2* 7, *d1* 6, *e1* 6, *f1* 8 and *h1* 8. The distances of setae were *c1-c1* 66, *c2*-*c2* 115, *d1*-*d1* 44, *e1*-*e1* 34, *f1*-*f1* 32, *h1*-*h1* 18, *c1*-*c2* 29, *c1*-*d1* 35, *c2*-*d1* 42, *d1*-*e1* 13, *e1*-*f1* 34 and *f1*-*h1* 31. 

*Cauda dorsum* (Figure 9). From the dorsal view, the posterior end of the idiodoma elongated significantly as a clearly defined long cauda, and gradually narrowed. It consisted of a caudal base with light striae, a caudal petiole with broad transverse plicated striae on the edge, and a smooth caudal xiphoid. The cauda was 226 long, measured from the posterior edge of the median shield to the end: the caudal base was 101, the caudal petiole was 74, and the caudal xiphoid was 51 in length. The anal region was dorsally located on the caudal base with fine dotted papillae, and the dorsal area of the genital region with *g3*–*g4* also had dotted papillae; the anal region had two pairs of pseudanal setae (*ps1*–*ps2*), 6 and 10 long, respectively; the length of the setae were *h2* 7, and lyrifissures (*ih*) were present.

*Venter* (Figure 10 and Figure 11). Coxae I–IV were fused, resulting in a clearly whole ventral shield completely covered with dotted papillae; coxae III–IV had an obviously reticulated pattern. One pair of propodogastral setae (*ppgs*) was close to coxae II, there were two pairs of hysterogastral setae (*hgs1*–*hgs2*), 6 and 14 in length, and there was one pair of clear foveolae medially located between the coxae III groups. The area between hysterogastral setae *hgs2* had transverse striation. The setal formula of coxal plates I–IV was 3-1-3-3 *sts*.

*Cauda venter* (Figure 10 and Figure 11). The cauda was 224 long: the caudal base was 99, including the genital area; the caudal petiole was broad with a plicated striae, 75 long; the caudal xiphoid was smooth and 50 long. The cauda was clearly defined and gradually narrowed, as in its dorsal view. The caudal base had horizontal striation except for the genital region (genital shield), with dotted papillae; the genital region had four pairs of genital setae (*g1*–*g4*), of which *g3*–*g4* were dorsally located (Figure 9), and two pairs of visible genital suckers. The lengths of setae *g1*–*g4* were 13, 14, 15 and 14, respectively. 

*Gnathosoma*. Palp (Figure 12A) were five-segmented, 112 long, with granulated papillae and terminating with a bifid claw. Palp chaetotaxy was as follows: trochanter without setae; basifemur with one dorsal simple seta; telofemur with one dorsal stout seta and one short and tapering apophysis; genu with two stout setae and two simple setae; and tibiotarsus with one stout seta, three simple setae and one acute solenidion. 

Chelicerae (Figure 12B): 92 long, with fine papillae; one cheliceral seta, 9 in length; it developed chela terminally.

Subcapitulum (Figure 12C): it had dotted papillae, 105 long, 70 wide, and two pairs of apophyses, of which one pair was smaller and claw-like and the other pair was blunt rod-like. It had two pairs of adoral setae (*ads1*, *ads2*), of which *ads1* was 6 and *ads2* was 4 in length, four pairs of hypostomal setae (*hg1*–*hg4*), where *hg2* and *hg4* were subequal in length and both were three times longer than *hg1* and *hg3*, and the lengths of *hg*-setae were *hg1* 12, *hg2* 29, *hg3* 10 and *hg4* 30. The distances between the bases of *hg*-setae were *hg1-hg1* 5, *hg2-hg2* 12, *hg3-hg3* 26, *hg4-hg4* 55, *hg1-hg2* 9, *hg2-hg3* 48 and *hg3-hg4* 19.

*Legs* (Figure 13). Leg IV was the longest and leg II was the shortest, with the tarsal lobes fairly well-developed; the dorsum of all leg segments had finely granulated papillae. The lengths of legs I–IV were 234, 220, 259 and 289. The lengths of tarsi I–IV were 84, 83, 100 and 101. Legs I–IV’s chaetotaxy was as follows: coxae I–IV 3-1-3-3 *sts*; trochanters I–IV 1-1-2-1 *sts*; basifemora I–IV: 4-4-3-0 *sts*; telofemora I–IV 4-4-4-4 *sts*; genua I–IV 2 *asl*, {1 *asl*, 1 long *bsl*, 1 *mst*}, 4 *sts*-1 *asl*, 1 long *bsl*, 5 *sts*-1 *asl*, 1 long *bsl*, 5 *sts*-1 *asl*, 1 long *bsl*, 5 *sts*; tibiae I–IV 1 long *bsl*, {1 *asl*, 1 *mst*}, 4 *sts*-1 *bsl*, 5 *sts*-1 *bsl*, 5 *sts*-1 *T*, 4 *sts*; and tarsi I–IV 3 *asl*, 1 long *bsl*, 1 *fam*, 1 *dtsl*, 16 *sts*-1 long *bsl*, 1 *dtsl*, 15 *sts*-1 *dtsl*, 14 *sts*-1 *dtsl* and 13 *sts*. 

Female and other developmental stages: unknown.

Remarks. This new species is similar to *C. mohanensis* Chen & Jin sp. nov., but differs from it in the following features: (1) *hg2* and *hg4* were subequal in length and both three times *hg1* and *hg3* (vs. *hg2* and *hg4* subequal in length and both four times *hg1* and *hg3* in *C. mohanensis* Chen & Jin sp. Nov.); (2) the chelicera reticulated pattern was absent (vs. present in *C. mohanensis* Chen & Jin sp. nov.); (3) the cauda was short, the caudal petiole was wider and the caudal xiphoid was shorter (vs. long and slender in *C. mohanensis* Chen & Jin sp. nov.).

Material examined. Holotype, male, collected from fallen leaves in Mohan Port (21°11′22.66″ N, 101°41′51.80″ E, elevation 893 m), Mengla County, Xishuangbanna Dai Autonomous Prefecture, Yunnan Province, China, on 20 November 2018, collector, Jianxin Chen and Xuesong Zhang. Slide No. YN-CU-201811201003. The slide is deposited in Institute of Entomology, Guizhou University, Guiyang, P. R. China (GUGC).

Etymology. The new specific epithet was formed by adding *neo*- (meaning new) to the name *C. mohanensis* Chen & Jin sp.nov., indicating its similarity to the latter species.

### 3.2. Brevicaudus Chen & Jin gen. nov.

Type species: *Brevicaudus trapezoides* Chen & Jin sp. nov.

Generic features (male): posterior of hysterosoma elongated with short conspicuous cauda, caudal petiole and caudal xiphoid short (as opposed to long in *Cunaxicaudus* Chen & Jin gen. nov.); palp between genu and tibiotarsus had one apophysis; the distance between setae *e1* and *d1* was approximately equal to that between *e1* and *f1; e1*, *c1* and *d1* were in a longitudinal line; lyrifissures (*im*) was situated lateral to *e1* and *f1*.

#### *Brevicaudus Trapezoides* Chen & Jin sp. nov.

Diagnosis. Setae *e1*, *f1* and *h1* were subequal and longer than the other dorsal setae (*lps*, *mps*, *c1*, *c2*, *d1*), with three pairs of hysterogastral setae (*hgs1*–*hgs3*); basifemora I–IV: 5-5-3-0 *sts*.

Description (Figure 14, Figure 15, Figure 16, Figure 17, Figure 18, Figure 19, Figure 20 and Figure 21); male (*n* = 4). 

The idiosoma length was 284 (283–315) from the base of the subcapitulum to the posterior edge of the hysterosomal shield, the width was 195 (190–220), and the posterior end of the hysterosoma was elongated as a short cauda (Figure 14, Figure 15 and Figure 16C). 

*Dorsum* (Figure 14, Figure 15 and Figure 16A,B). The propodosomal and hysterosomal shields were entirely complemented by reticulations, integument with striae. The propodosomal shield was 70 (70–85) long and 118 (95–130) wide, sclerotized and with a reticulated pattern, bearing one pair of anterior (*at*) and one pair of posterior (*pt*) setose trichobothria and two pairs of tactile setae (*lps* and *mps*); *at* was shorter than *pt*, *lps* was nearer to *pt* than to *at*; the area was anterior to setae *at* papillary. The setal lengths and distances of setae were *at* 175 (190–200), *pt* 259 (242–265), *lps* 10 (7–10), *mps* 12 (9–10); *at-at* 20 (20–23), *pt-pt* 125 (109–133), *lps-lps* 110 (96–124), *mps-mps* 53 (45–60), *lps-mps* 43 (31–42), *at-lps* 68 (69–75), *pt-mps* 37 (30–36), *pt-lps* 20 (10–22), *at-mps* 74 (75–88) and *at-pt* 82 (84–93). 

The hysterosomal median shield was 121 (118–126) long and 126 (117–142) wide, with five pairs of dorsal simple setae (*c1*, *c2*, *d1*, *e1*, *f1*) and one pair of lyrifissures (*im*), which was situated laterally to *e1* and *f1*. The distance between setae *e1* and *d1* was about equal to that between *e1* and *f1; e1*, *c1* and *d1* were in a longitudinal line. *e1*, *f1* and *h1* were subequal and longer than *lps*, *mps*, *c1*, *c2* and *d1*. Setae *h1* was situated on the striated integument of the caudal base and was equal to *e1*, *f1* in length. The lengths of six pairs of dorsal setae were *c1* 12 (9–11), *c2* 10 (8–11), *d1* 12 (8–11), *e1* 23 (13–19), *f1* 18 (13–19), *h1* 21 (16–20). Distances of setae: *c1-c1* 39 (35–45), *c2*-*c2* 111 (103–114), *d1*-*d1* 42 (31–42), *e1*-*e1* 44 (24–46), *f1*-*f1* 26 (24–33), *h1*-*h1* 22 (16–26), *c1*-*c2* 47 (41–46), *c1*-*d1* 36 (32–37), *c2*-*d1* 36 (31–37), *d1*-*e1* 37 (34–36), *e1*-*f1* 29 (27–31) and *f1*-*h1* 30 (24–39). 

*Cauda dorsum* (Figure 14, Figure 15 and Figure 16C). The posterior end of the idiodoma terminated into a distinctly developed cauda, which consisted of a caudal base with light striae and an approximately trapezoid (posteriorly widened) caudal petiole and caudal xiphoid, together almost as long as the caudal base, sharply reduced from the end of the caudal petiole. The cauda length was 100 (85–117): the caudal base was 50 (40–65), the caudal petiole was 30 (25–35) and the caudal xiphoid was 20 (18–22). The anal-genital region on the caudal base had fine dotted papillae; the genital region was *g3*–*g4*; the anal region had two pairs of pseudanal setae (*ps1*–*ps2*), 10 (12–14) and 11 (11–12) in length, respectively, and one pair of *h2* was 10 (8–11) in length and one pair was lyrifissures (*ih*).

*Venter* (Figure 17 and Figure 18A,B). Coxae I–IV fused, forming a clearly whole ventral shield completely covered with dotted papillae; coxae III–IV had an obvious reticulated pattern. One pair of propodogastral setae (*ppgs*), 10 (10–12), were close to coxae II; three pairs of hysterogastral setae (*hgs1*–*hgs3*) had lengths of setae *hgs1*–*hgs3*: 10 (10–16), 13 (10–11), 17 (11–17); one pair of clear foveolae were medially located between the coxae III groups. Arae between *hgs2* and the posterior edge of the median shield had horizontal striation. The setal formula of coxal plates I–IV was 3-1-3-3 *sts*. 

*Cauda venter* (Figure 17). The cauda was clearly defined as in its dorsal view, clearly separated from the ventral shield. The cauda was 110 (85–117) long: the caudal base was 55 (40–65) long, including the genital area; the caudal petiole was smooth and 35 (25–30) long; the caudal xiphoid was smooth and 20 (18–22) long. The caudal base had horizontal striation, except the genital area (genital shield); the genital shield had dotted papillae, two visible pairs of genital suckers, and four pairs of genital setae (*g1*–*g4*), of which *g3*–*g4* were dorsally located (Figure 15). The lengths of setae *g1*– *g4* were 14 (15–16), 14 (15–19), 18 (15–20) and 15 (15–17), respectively. 

*Gnathosoma*. Palp (Figure 19A and Figure 20A): five-segmented, 155 (145–169) long, with granulated papillae and terminating with a claw. Palp chaetotaxy was as follows: trochanter without setae; basifemur with one dorsal simple seta; telofemur with one dorsal stout seta and one short and tapering apophysis; genu with three stout setae, one simple seta and an apophysis close to base of tibiotarsus; and tibiotarsus with one stout seta, three simple setae and one terminating solenidion. 

Chelicerae (Figure 19A and Figure 20B): 125 (115–126) long, with fine papillae; the length of the cheliceral seta was 14 (13–14), which was located approximately 103 (95–106) from the base; one developed chela terminally.

Subcapitulum (Figure 19B and Figure 20C): dotted papillae 132 (113–137) long and 63 (70–90) wide, with two pairs of apophyses, of which one pair was smaller and claw-like and the other was blunt and rod-like. There were two pairs of adoral setae (*ads1*, *ads2*), of which *ads1* was 11 (8–9) and *ads2* was 5 (4–5) in length, and four pairs of hypostomal setae (*hg1*–*hg4*). The lengths of *hg*-setae were *hg1* 16 (14–17), *hg2* 21 (23–24), *hg3* 8 (7–8) and *hg4* 31 (30–37). Distances between bases of *hg*-setae: *hg1-hg1* 6 (5–6), *hg2-hg2* 14 (13–17), *hg3-hg3* 30 (25–30), *hg4-hg4* 58 (59–69), *hg1-hg2* 22 (19–25), *hg2-hg3* 53 (47–65) and *hg3-hg4* 25 (20–24).

*Legs* (Figure 21). Leg IV was the longest and leg II was the shortest, and the tarsal lobes were well-developed. The lengths of legs I–IV were 270 (237–289), 230 (207–256), 265 (239–286), 288 (260–293). Lengths of tarsi I–IV: 110 (97–120), 99 (78–100), 97 (94–118) and 98 (93–118). Leg I–IV’s chaetotaxy: coxae I–IV 3-1-3-3 *sts*; trochanters I–IV 1-1-2-1 *sts*; basifemora I–IV: 5-5-3-0 *sts*; telofemora I–IV 4-4-4-4 *sts*; genua I–IV 2 *asl*, {1 *asl*, 1 long *bsl*, 1 *mst*}, 4 *sts*-1 long *bsl*, 1 *asl*, 5 *sts*-1 *asl*, 5 *sts*-2 *asl*, 5 *sts*; tibiae I–IV 1 long *bsl*, {1 *asl*, 1 *mst*}, 4 *sts*-1 *asl*, 5 *sts*-1 *bsl*, 5 *sts*-1 *T*, 4 *sts*; and tarsi I–IV 3 *asl*, 1 long *bsl*, 1 *fam*, 1 *dtsl*, 17 *sts*-1 long *bsl*, 1 *dtsl*, 17 *sts*-1 *dtsl*, 13 *sts*-1 *dtsl* and 13 *sts*. 

Female and other developmental stages. unknown.

Remarks. the new species resembles *C. mohanensis* Chen & Jin sp. nov. and *C. neomohanensis* Chen & Jin sp. nov., but significantly differs from the latter two by generic features.

Material examined. Holotype, male, collected from fallen leaves in Mohan Port (21°11′22.66″ N, 101°41′51.80″ E, elevation 893 m), Mengla County, Xishuangbanna Dai Autonomous Prefecture, Yunnan Province, China, on 9 June 2019, collector, Jianxin Chen, slide No. YN-CU-201906090208. Paratypes, two males, were collected from fallen leaves in Mohan Port (21°11′22.66″ N, 101°41′51.80″ E, elevation 893 m), Mengla County, Xishuangbanna Dai Autonomous Prefecture, Yunnan Province, China, on 6 June 2019, by Jianxin Chen; slides No. YN-CU-201906060101, YN-CU-201906060207. Paratype, one male, was collected from fallen leaves in Mohan Port (21°11′22.66″ N, 101°41′51.80″ E, elevation 893 m), Mengla County, Xishuangbanna Dai Autonomous Prefecture, Yunnan Province, China, on 8 June 2019, collector, Jianxin Chen; slide No. YN-CU-201906080201. All types of materials were deposited at the Institute of Entomology, Guizhou University, Guiyang, P. R. China (GUGC).

Etymology. Latin word ‘brevis’ means short in general. This refers to the short cauda of the new genus, which is much longer in *Cunaxicaudus* Chen & Jin gen. nov.; the new species is named for the caudal lobe, which is approximately trapezoid.

## 4. Discussion

According to Simely [5] and Skvarla et al. [10], the number of segments and length of palp, absence or presence of multi-branched seta on palp III (telofemur), cheliceral seta absent or present and *T* absent or present on tibiae IV were used as the main diagnostic features for the subfamilies in Cunaxidae. For example: palp with three segments = Counaxoidinae; palp with four segments = Scirulinae; palp with five segments and multi-branched seta present on palp III (telofemur) = Bonzinae; palp with five segments and multi-branched seta absent on palp III (telofemur), and palp extending beyond the subcapitulum by at least the distal half of palp IV(telofemur) = Cunaxinae; palp with five segments, multi-branched seta absent on palp III (telofemur), and palp not extending beyond the subcapitulum by more than the distal half of palp IV (genua), *T* present on tibiae IV, cheliceral seta usually present = Coleoscirinae; palp with five segments, multi-branched seta absent on palp III (telofemur), and palp not extending beyond the subcapitulum by more than the distal half of palp IV(genua), *T* absent on tibiae IV, and cheliceral seta absent, seta *hg1* geniculate = Orangescirulinae. 

Our specimens of the three proposed new species have the following characteristics: palp with five segments, multi-branched seta absent on palp III (telofemur), and palp extending beyond the subcapitulum by at least the distal half of the palp IV (genua), cheliceral seta present and *T* present on tibiae IV. Moreover, we think that the feature of posterior configuration of hysterosoma can also be used as a main diagnostic feature for the subfamilies in the family Cunaxidae. The posterior end of the idiosoma elongated remarkably to form a cauda-like structure, which is an unusual new trait unknown in the family to date.

There was a tendency for the posterior area to the hysterosomal shield, including the genital region and anal plate (region) in the ventral view, to have the idiosoma elongated rearward in some members of Cunaxidae, in which case the anal pore was located on the end of the idiosoma and the anal plate (region) covered both ventral and dorsal around the pore in both males and females. What is more significant than that is that the end of the idiosoma obviously extended with the exposed aedeagus in males of some species, such as *Dactyloscirus humuli* (illustrated and mentioned) [26] and *Armascirus hastus* (illustrated but not mentioned) [27]. However, the completed and fully equipped cauda-like device is primarily defined in the present study.

A similar structure, with cauda and petiole as mating apparatuses [28], presents in some males of the water mite family Arrenuridae, especially those of the subfamily Arrenurinae, which makes the arrenurid mites sexual dimorphic [29]. The Arrenurid cauda is formed by an elongation of the hysterosoma posteriorly to a variable extent, and the petiole, either simple or complicated, and is a device derived from the integument of the idiosoma either caudally or directly from the posterior edge of dorsal shield [30]. The genital field or anal area may be or not on the cauda. Jin et al. [31] hypothesized that the formation and evolution of the arrenurid cauda and petiole was driven by the behavioral evolution of the reproductive mechanism from non-mating (indirect sperm transfer) to mating (direct sperm transfer). The cauda of Cunaxicaudinae Chen & Jin subfam. nov. is highly homologous, but remarkably different from that of the arrenurid mites. In cunaxicaud mites, both the genital area and the anal area were located on the male cauda, which implies that the traits (idiosoma posteriorly elongated with the accessory devices of cunaxicaud and arrenurid) evolved independently in two different branches. Interestingly, the aedeagus was not observed internally in all specimens of Cunaxicaudinae Chen & Jin subfam. nov., and therefore we propose that the caudal xiphoid might be the evaginated aedeagus with the idiosoma extending. Such an aedeagus was also observed in *Dactyloscirus condyles* [32].

Unfortunately, the females of Cunaxicaudinae Chen & Jin subfam. nov. were not collected. According to the literature, there are certain sexual differences, although they are not regarded as typical sexual dimorphism in the known members of Cunaxidae. These include the number of dorsal shields (females only have one, while males have two) [5,27,33,34], the setal formula of coxal plates (the setal formula of coxal plates I–IV was 3-2-3-3 *sts* in females, while 3-1-3-3 *sts* in males) [10], leg chaetotaxy (generally, males have longer *bsl* than females, especially on genua I–IV, tibia I and tarsi I–II) [35] and the number of genital setae (*g*) and eugenital setae (*eu*) (female genital setae were more than male, females had a pair of eugenital setae present, while in males this was absent) [13,16].

Therefore, we could put forward the hypothesis that the cauda of the new taxa may be the result of the evolution of the reproductive mode from indirect sperm transfer to direct sperm transfer; the new taxa may have sexual dimorphism, just like arrenurid mites, with the males having an exceptional body consisting of the mating device and females having a common oval body.

## 5. Conclusions

In the present work, a new subfamily, Cunaxicaudinae Chen & Jin subfam. nov., was described and illustrated based on a unique cauda-like structure in male Cunaxicaudinae. Additionally, we defined and discussed its morphology from the functional perspective. The findings highlight the species diversity and morphological evolution trends of the Cunaxidae.

## 6. Key to Subfamilies of Cunaxidae (Modified from Skvarla et al.) [10]

1. Hysterosoma cauda structure present in males (Figure 2, Figure 3B, Figure 4, Figure 5A,B, Figure 9, Figure 10, Figure 11, Figure 14, Figure 15, Figure 16C and Figure 17)
**Cunaxicaudinae Chen & Jin subfam. nov.**
-Hysterosoma cauda structure absent in males22. Palp telofemoral multi-branched seta present (except *Parabonzia*)
**Bonziinae Den Heyer, 1978**
-Palp telofemoral multi-branched seta absent33. Palp three-segmented
**Cunaxoidinae Den Heyer, 1978**
-Palp four- or five-segmented44. Palp four-segmented
**Scirulinae Den Heyer, 1980**
-Palp five-segmented55. Palp extended beyond the subcapitulum by at least the distal half of the genua

**Cunaxinae Den Heyer, 1978**
-Palp not extended beyond the subcapitulum by more than the distal half of the genua66. *T* on tibiae IV present; cheliceral seta usually present; seta *hg1* not geniculate
**Coleoscirinae Den Heyer, 1978**
-*T* on tibiae IV absent; cheliceral seta absent; seta *hg1* geniculate
**Orangescirulinae Bu & Li, 1987**


## Figures and Tables

**Figure 1 animals-13-01363-f001:**
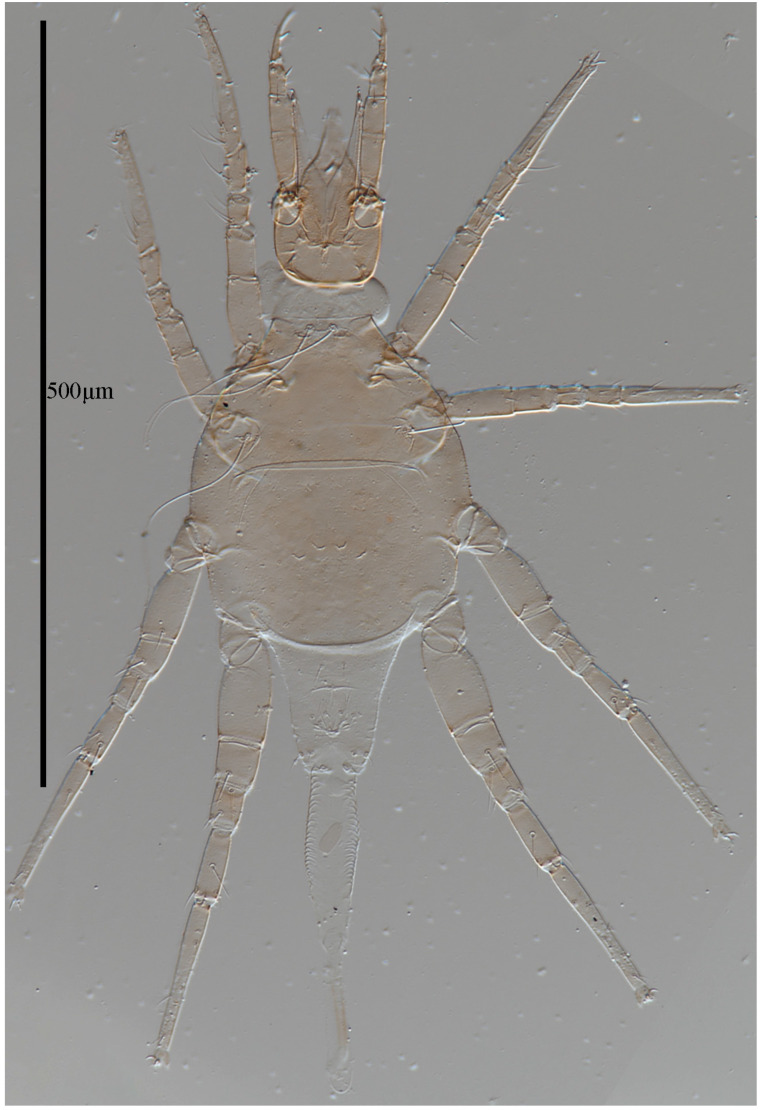
*Cunaxicaudus mohanensis* sp. nov. (male). Entire specimen (photo). Scale bar = 500 μm.

**Figure 2 animals-13-01363-f002:**
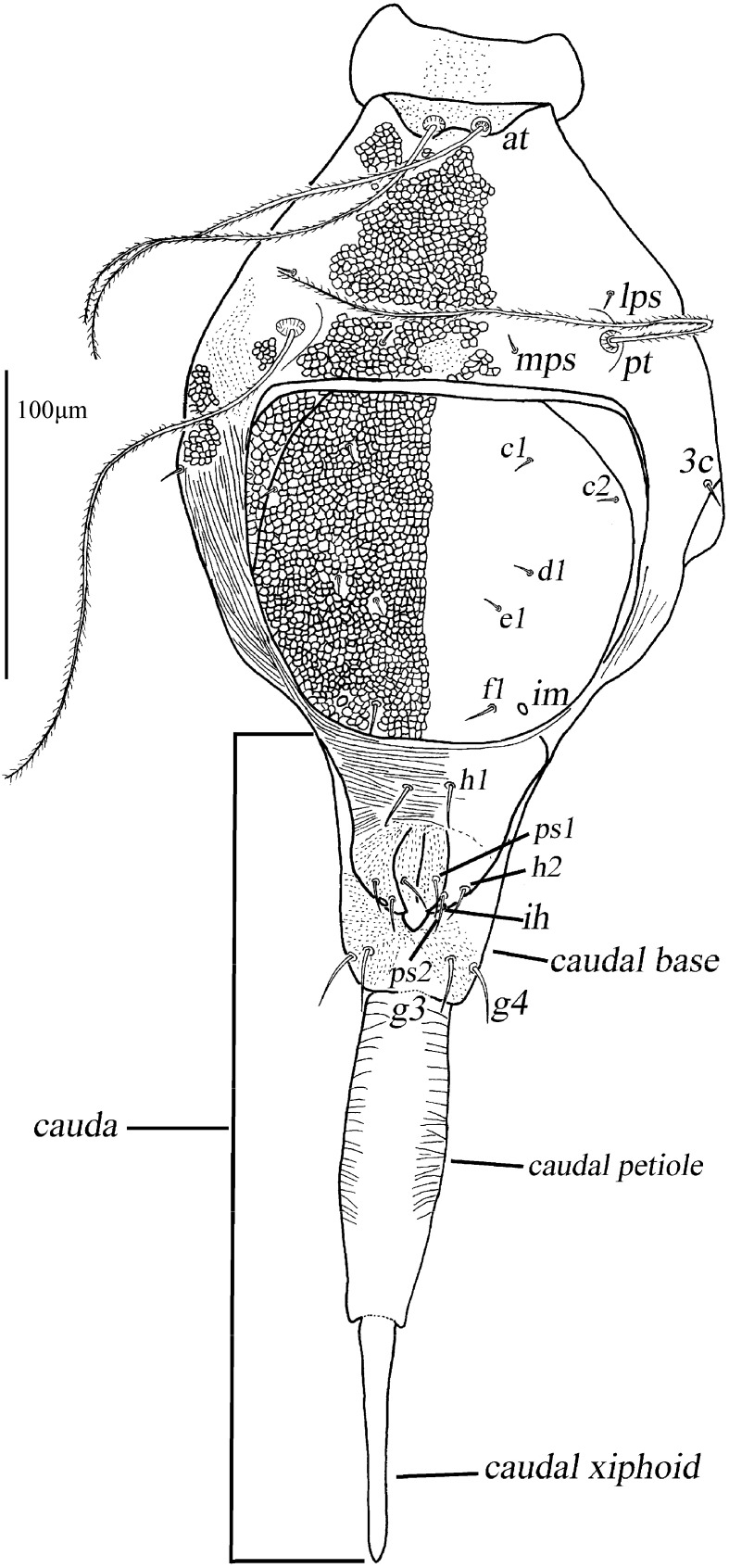
*Cunaxicaudus mohanensis* sp. nov. (male). Dorsal idiosoma. Scale bar = 100 μm.

**Figure 3 animals-13-01363-f003:**
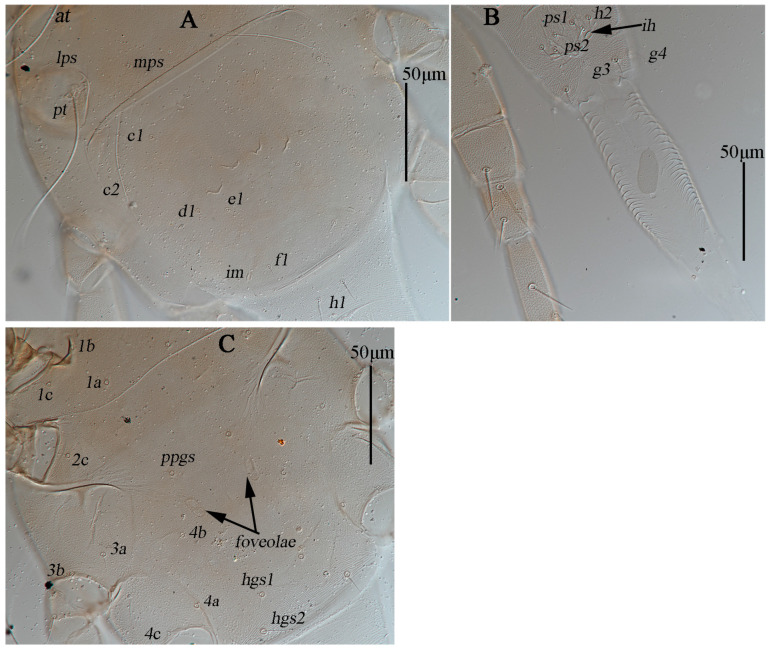
*Cunaxicaudus mohanensis* sp. nov. (male). (**A**,**B**)—Dorsal idiosoma (photo). (**C**)—Ventral idiosoma (photo). Scale bar = 50 μm.

**Figure 4 animals-13-01363-f004:**
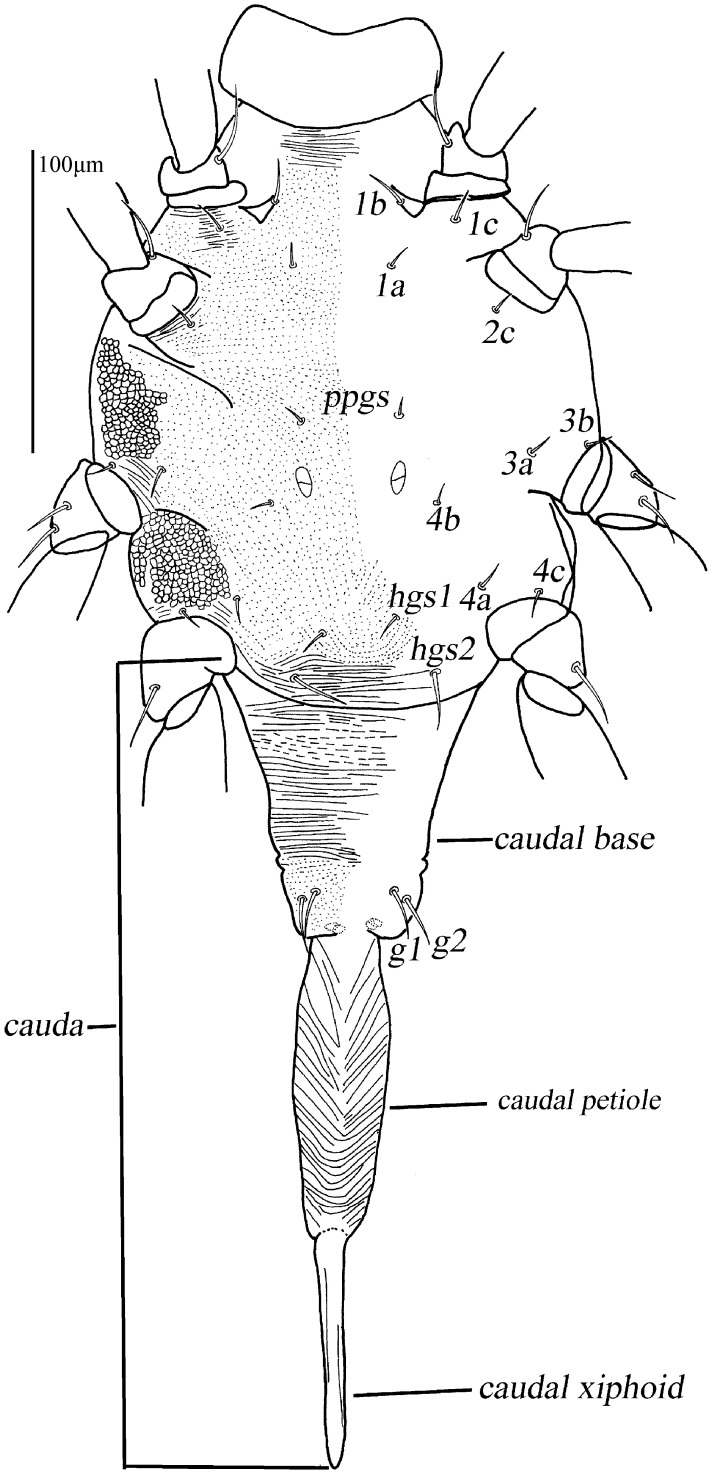
*Cunaxicaudus mohanensis* sp. nov. (male). Ventral idiosoma. Scale bar = 100 μm.

**Figure 5 animals-13-01363-f005:**
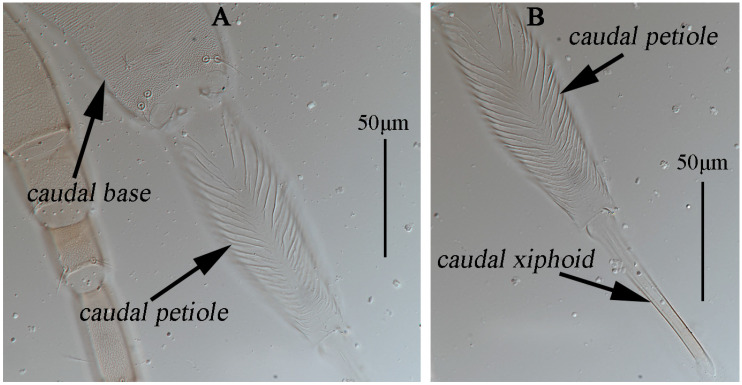
*Cunaxicaudus mohanensis* sp. nov. (male). (**A**)—Ventral caudal base and petiole (photo). (**B**)—Ventral caudal petiole and xiphoid (photo). Scale bar = 50 μm.

**Figure 6 animals-13-01363-f006:**
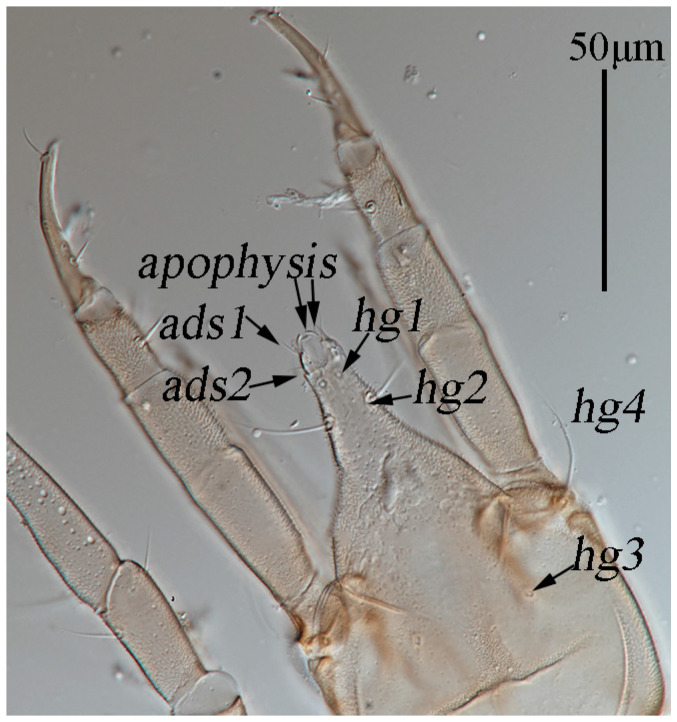
*Cunaxicaudus mohanensis* sp. nov. (male). Gnathosoma (photo). Scale bar = 50 μm.

**Figure 7 animals-13-01363-f007:**
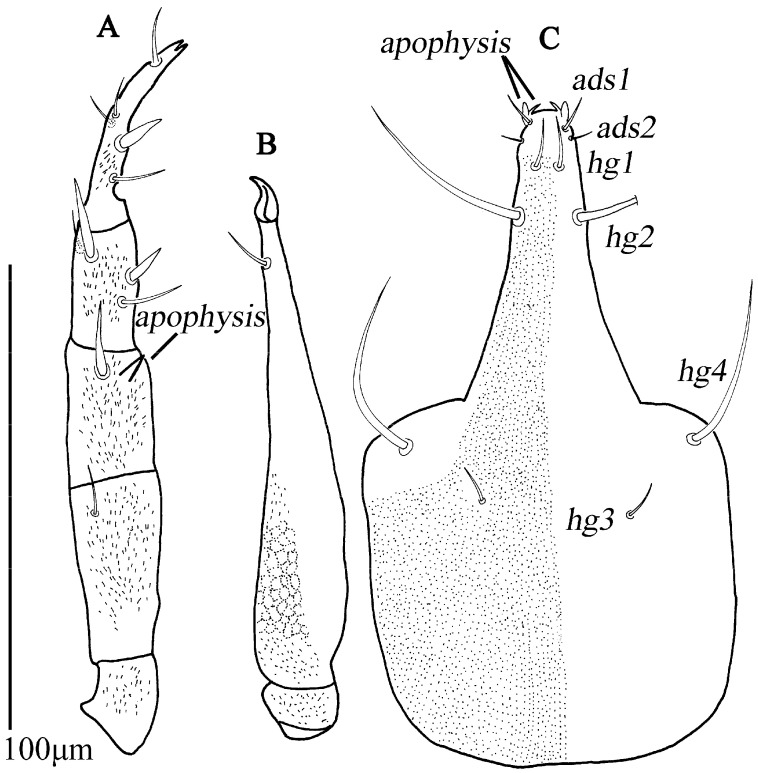
*Cunaxicaudus mohanensis* sp. nov. (male). (**A**)—Palp. (**B**)—Chelicerae. (**C**)—Subcapitulum. Scale bar = 100 μm.

**Figure 8 animals-13-01363-f008:**
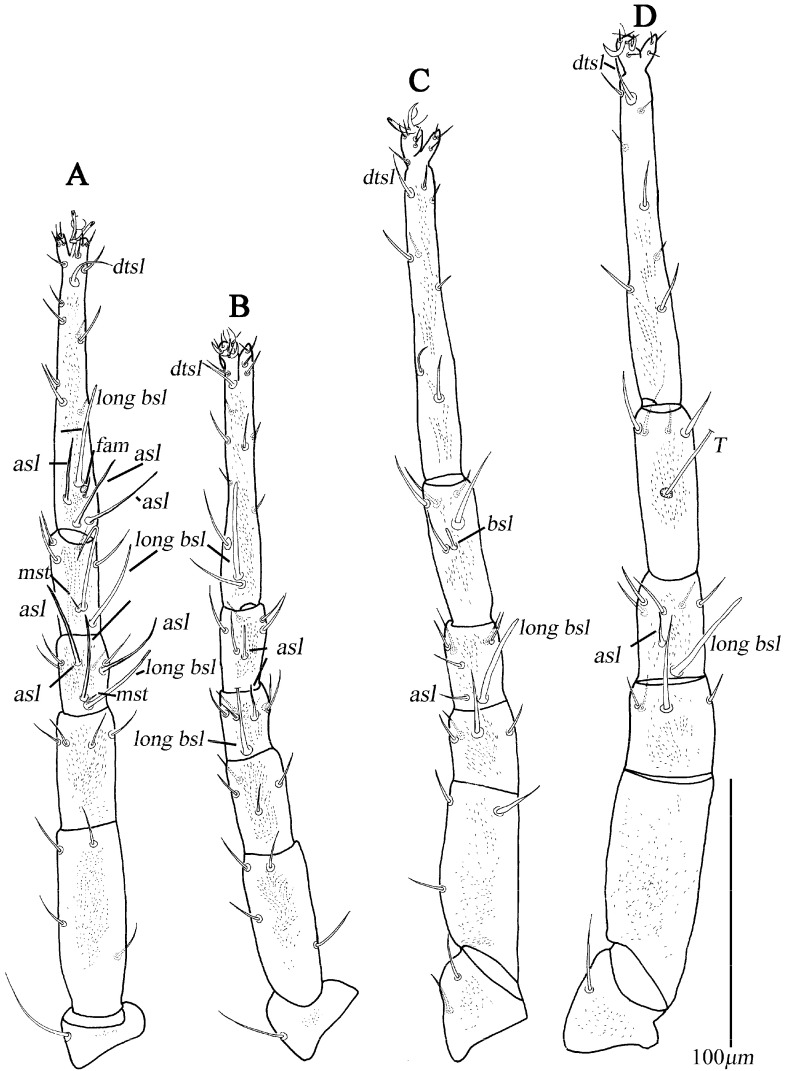
*Cunaxicaudus mohanensis* sp. nov. (male). (**A**–**D**)—Legs I–IV, respectively. Scale bar = 100 μm.

**Figure 9 animals-13-01363-f009:**
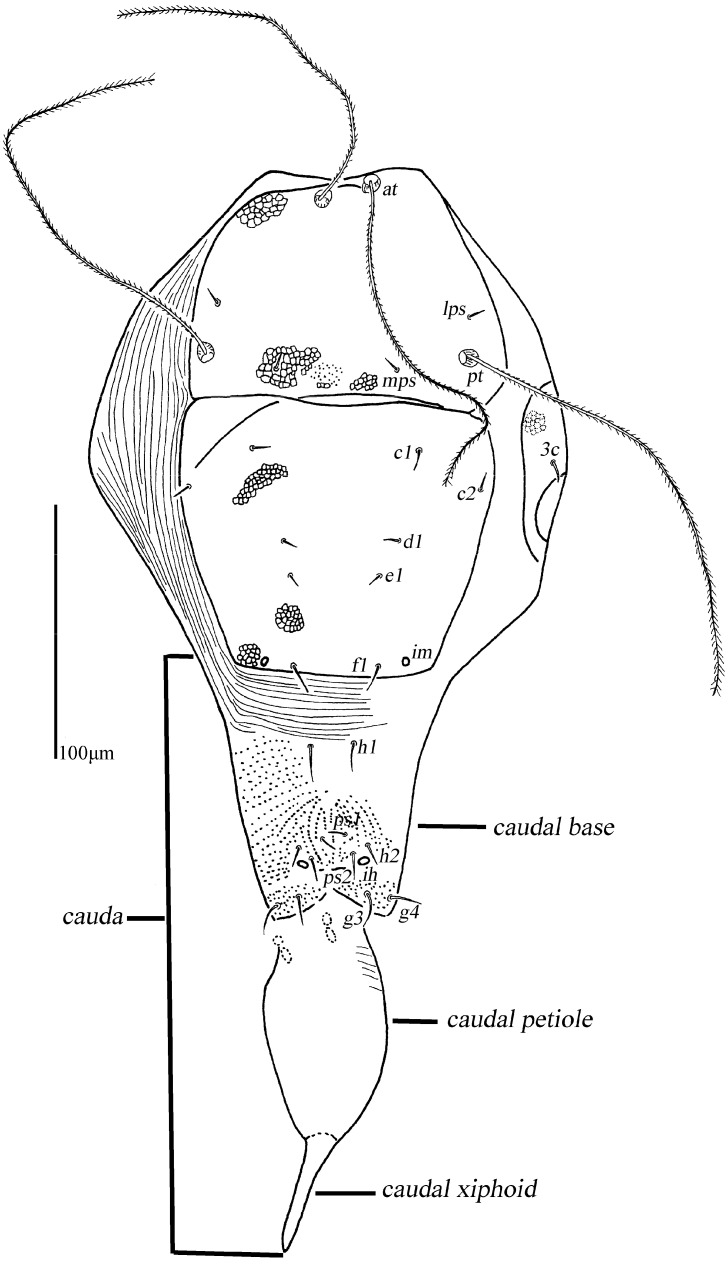
*Cunaxicaudus neomohanensis* sp. nov. (male). Dorsal idiosoma. Scale bar = 100 μm.

**Figure 10 animals-13-01363-f010:**
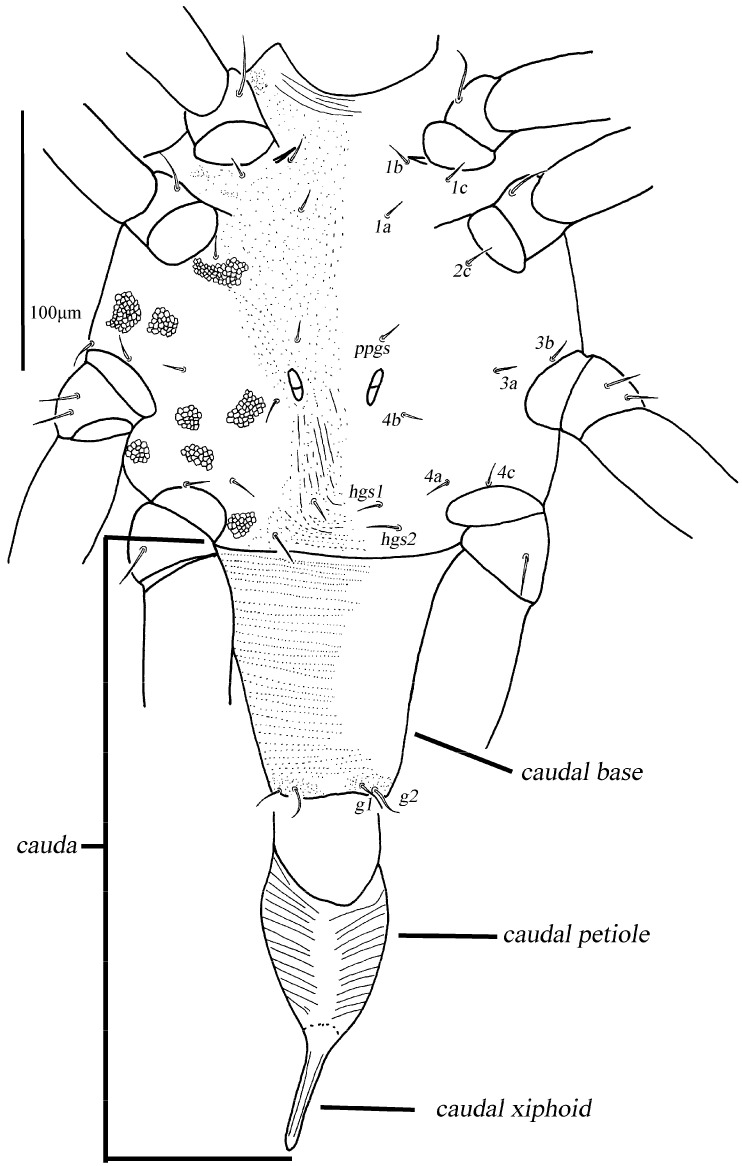
*Cunaxicaudus neomohanensis* sp. nov. (male). Ventral idiosoma. Scale bar = 100 μm.

**Figure 11 animals-13-01363-f011:**
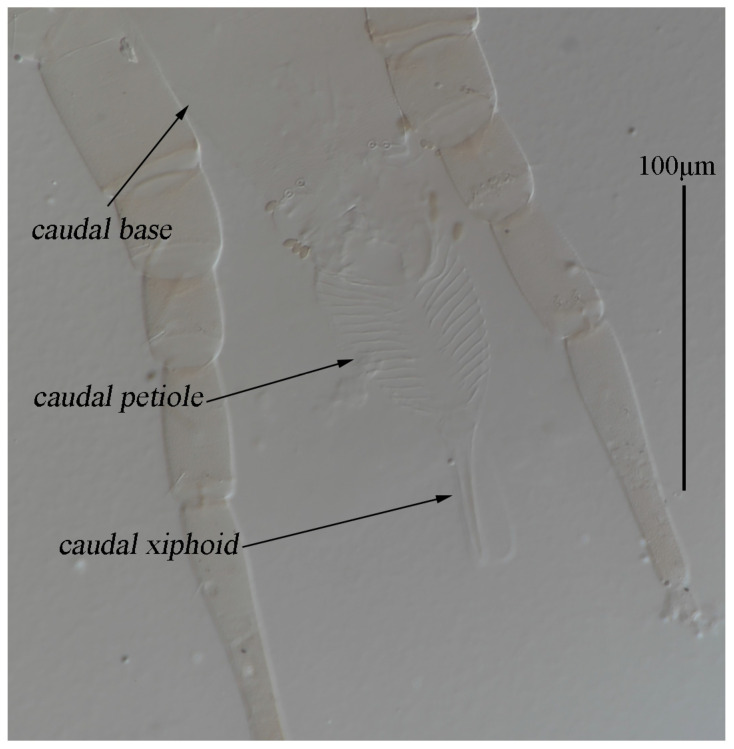
*Cunaxicaudus neomohanensis* sp. nov. (male). Cauda (photo). Scale bar = 100 μm.

**Figure 12 animals-13-01363-f012:**
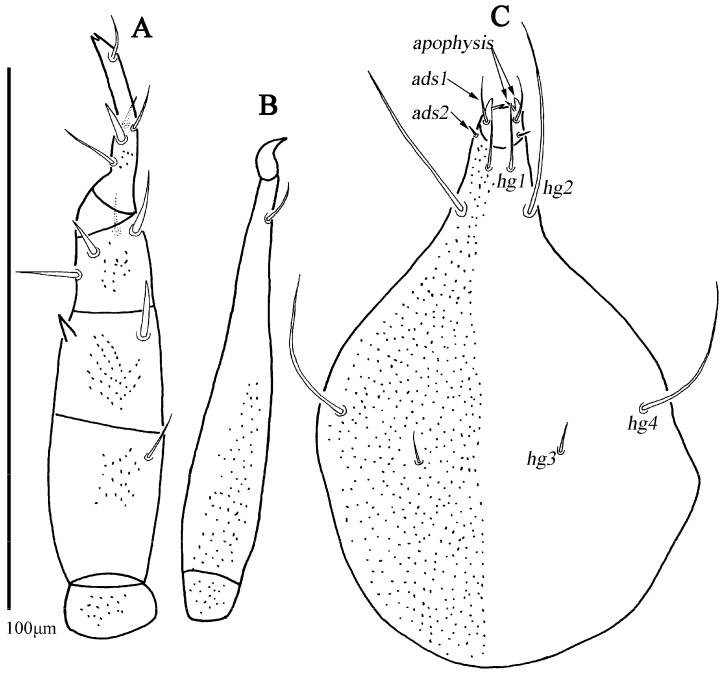
*Cunaxicaudus neomohanensis* sp. nov. (male). (**A**)—Palp. (**B**)—Chelicerae. (**C**)—Subcapitulum. Scale bar = 100 μm.

**Figure 13 animals-13-01363-f013:**
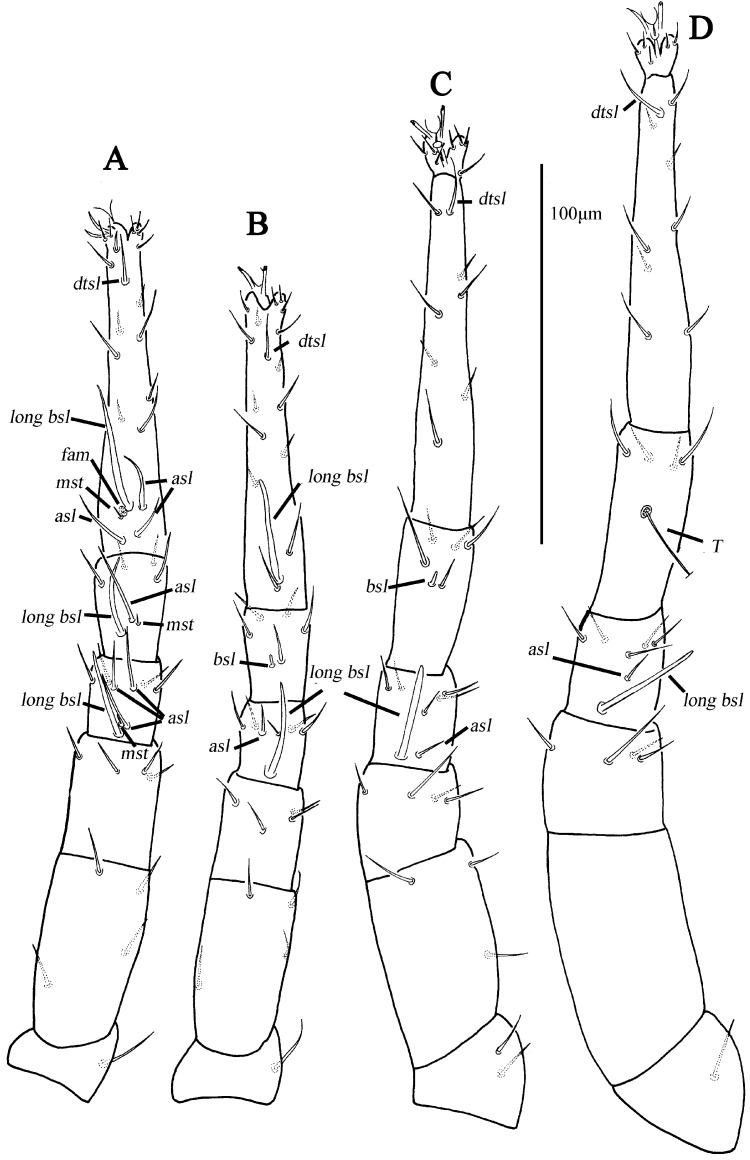
*Cunaxicaudus neomohanensis* sp. nov. (male). (**A**–**D**)—Legs I–IV, respectively. Scale bar = 100 μm.

**Figure 14 animals-13-01363-f014:**
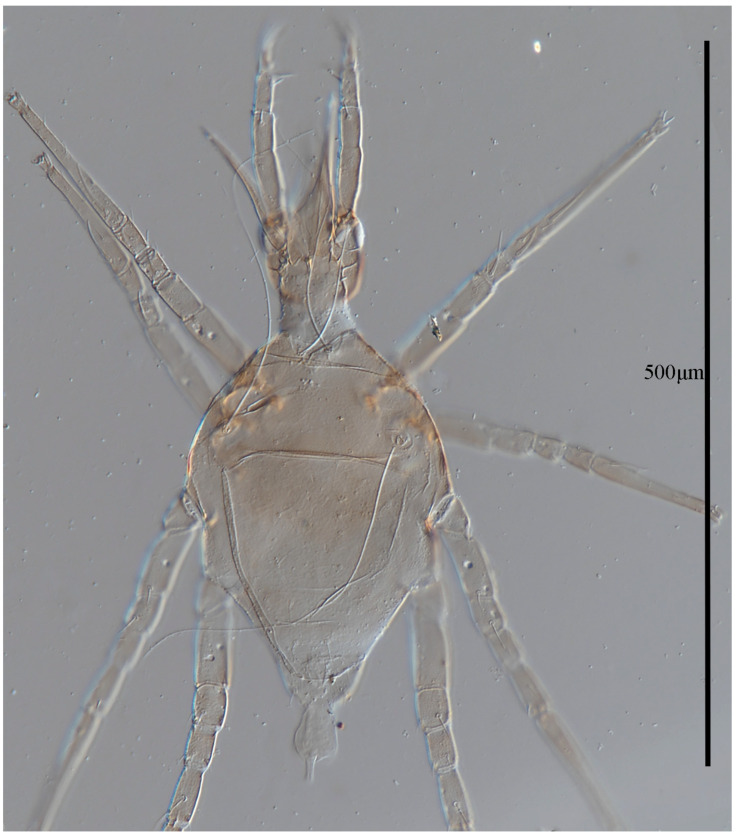
*Brevicaudus trapezoides* sp. nov. (male). Entire specimen (photo). Scale bar = 500 μm.

**Figure 15 animals-13-01363-f015:**
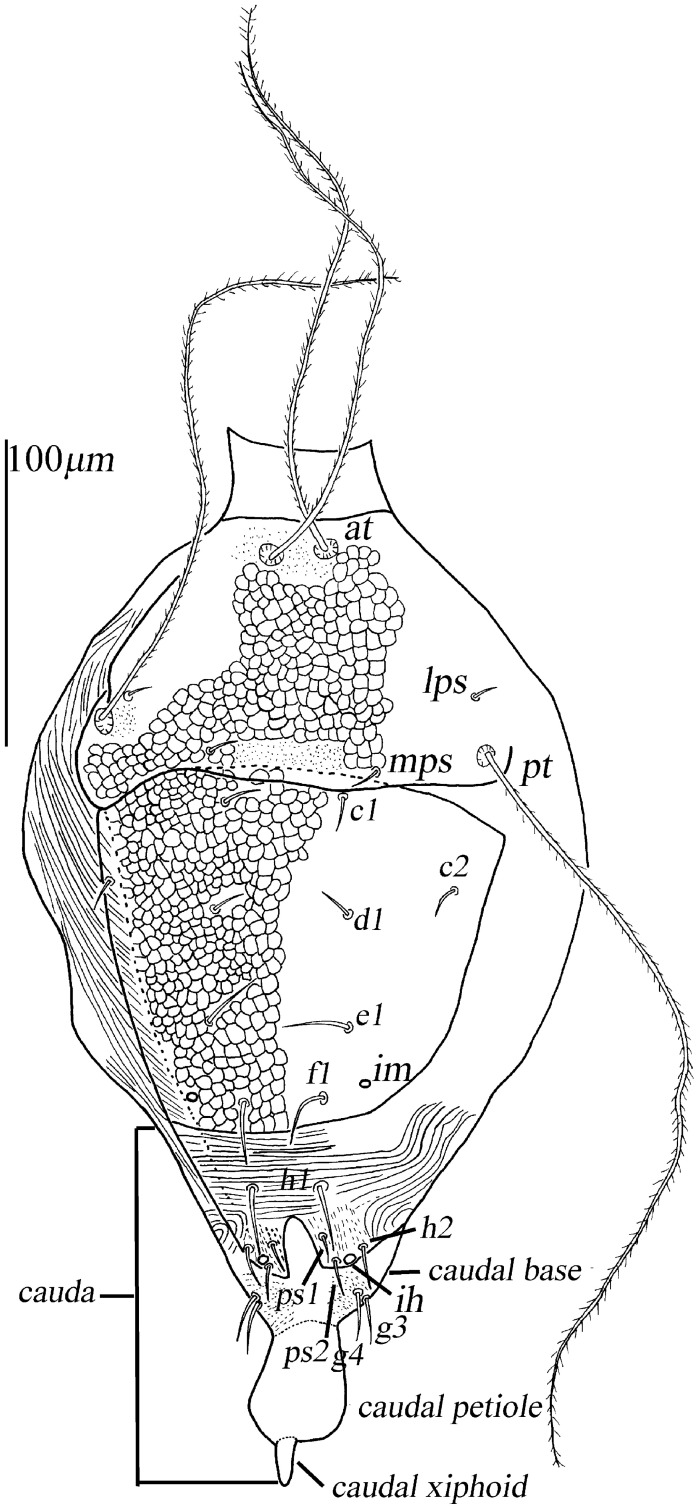
*Brevicaudus trapezoides* sp. nov. (male). Dorsal idiosoma. Scale bar = 100 μm.

**Figure 16 animals-13-01363-f016:**
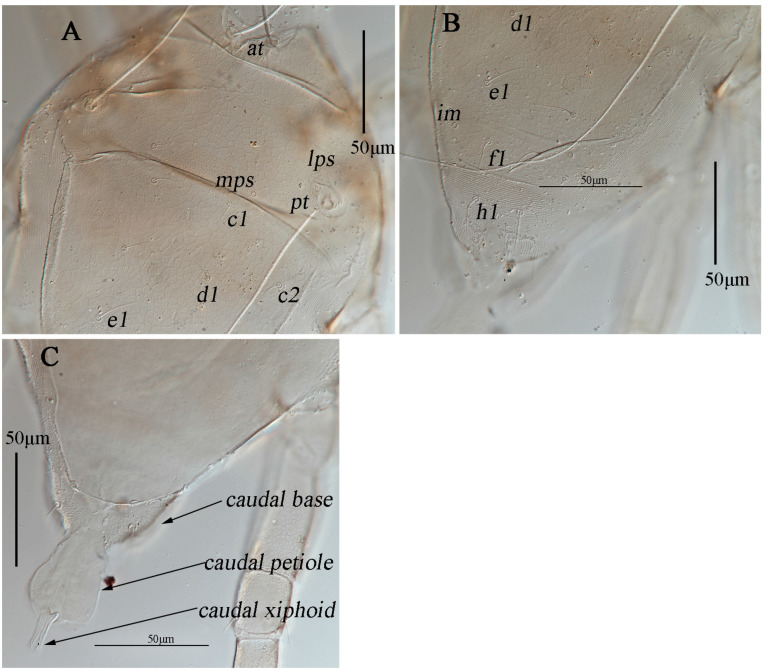
*Brevicaudus trapezoides* sp. nov. (male). (**A**,**B**)—Dorsal idiosoma (photo). (**C**)—Cauda (photo). Scale bar = 50 μm.

**Figure 17 animals-13-01363-f017:**
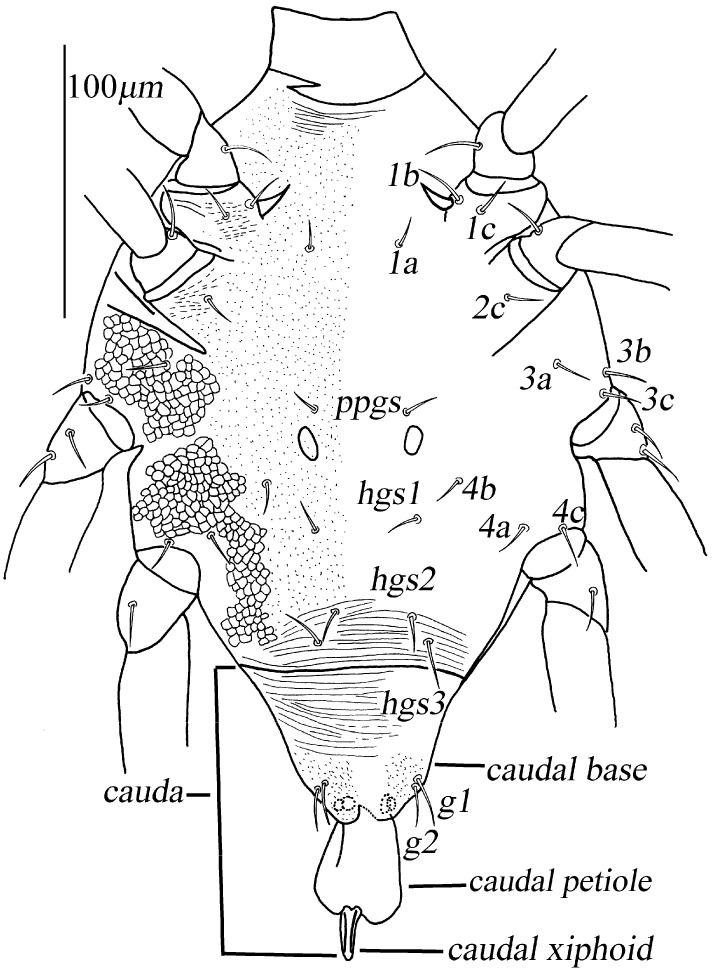
*Brevicaudus trapezoides* sp. nov. (male). Ventral idiosoma. Scale bar = 100 μm.

**Figure 18 animals-13-01363-f018:**
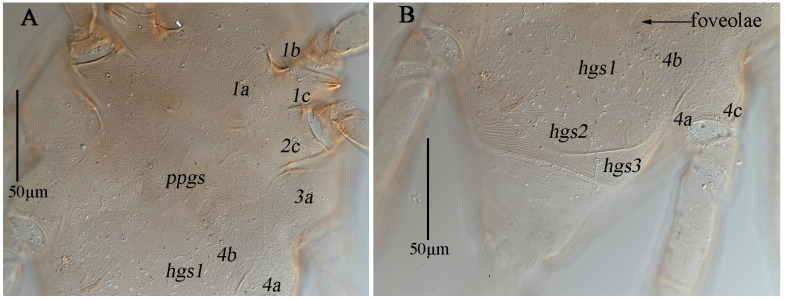
*Brevicaudus trapezoides* sp. nov. (male). (**A**,**B**)—Ventral idiosoma (photo). Scale bar = 50 μm.

**Figure 19 animals-13-01363-f019:**
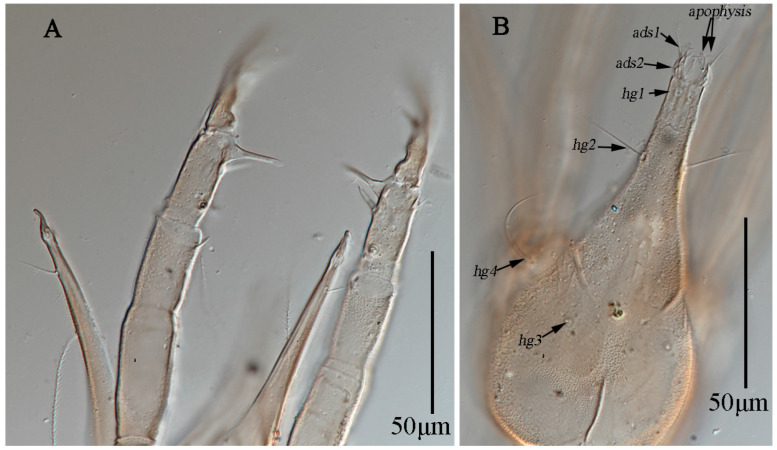
*Brevicaudus trapezoides* sp. nov. (male). (**A**)—Palp and chelicerae (photo). (**B**)—Subcapitulum (photo). Scale bar = 50 μm.

**Figure 20 animals-13-01363-f020:**
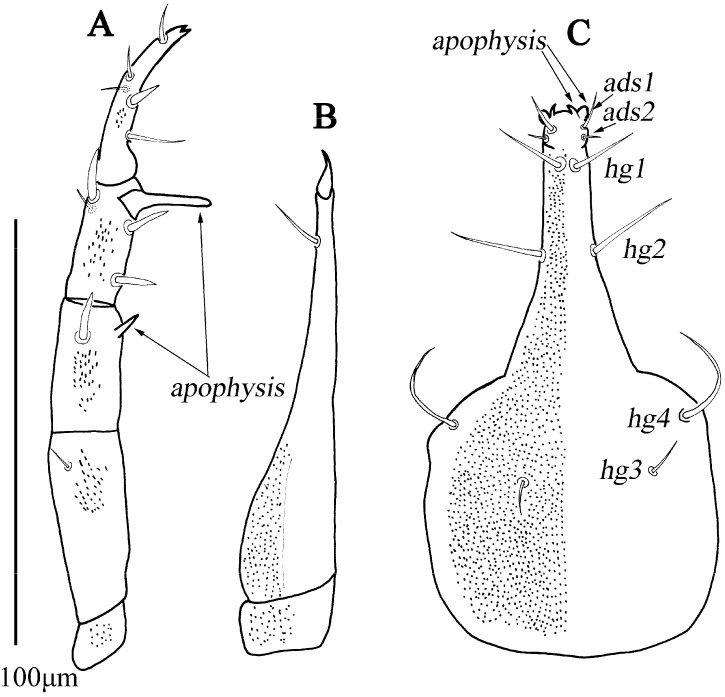
*Brevicaudus trapezoides* sp. nov. (male). (**A**)—Palp. (**B**)—Chelicerae. (**C**)—Subcapitulum. Scale bar = 100 μm.

**Figure 21 animals-13-01363-f021:**
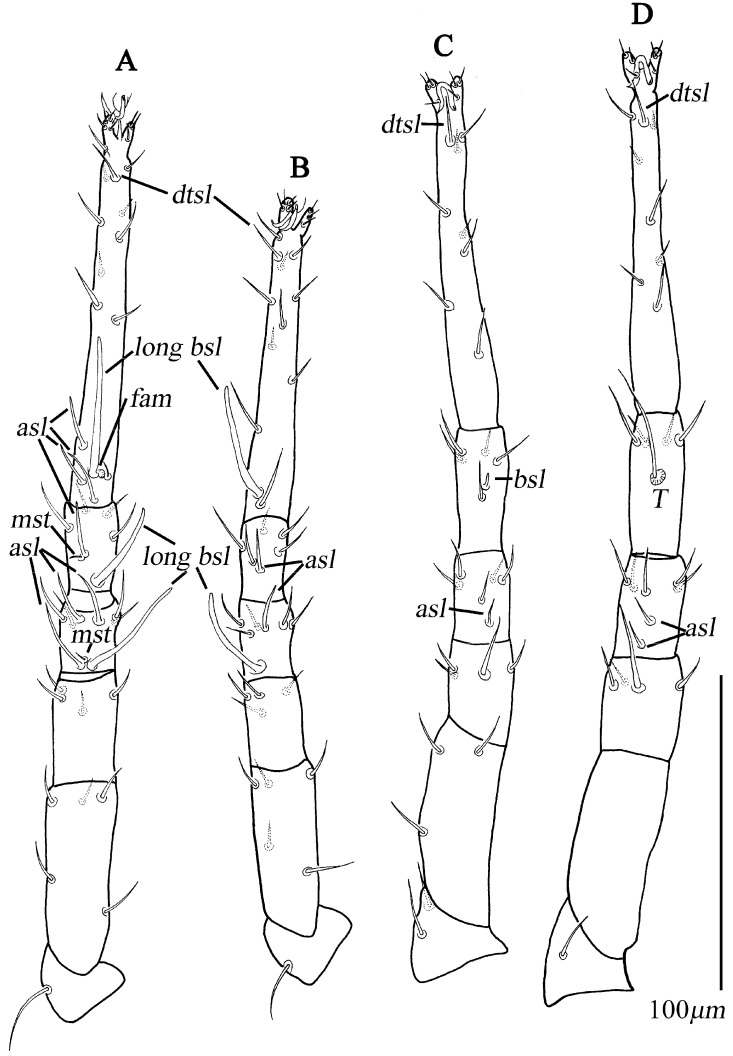
*Brevicaudus trapezoides* sp. nov. (Male). (**A**–**D**)—Legs I–IV, respectively. Scale bar = 100 μm.

## Data Availability

The data presented in this study are available on request from the corresponding author.

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
