# Peer review of "The Unique Cauda-Liked Structure Represents a New Subfamily of Cunaxidae: Description of New Taxa and Discussion on Functional Morphologyâ€"

_animals, 2023, doi:10.3390/ani13081363_

Round 1
Reviewer 1 Report (Previous Reviewer 1)
I have only one more comment. It is extremely surprising that females of the new species were not collected. The authors could add a few suggestions about why this happened, at the end of the Discussion.
Author Response
Amphigenesis and parthenogenesis exist in Cunaxidae. Generally, most of the collected mites are full of females and few are males. The collection is accidental. As a result, specific proposals are unclear.
Reviewer 2 Report (New Reviewer)
The manuscript proposed a new subfamily, two new genera and three new species based on males with a cauda-liked structure. Upon reviewing the material, I find it to be well written and supported by high-quality figures and images.
However, I have observed a significant misconception regarding the definition of the new subfamily, which suggests a lack of understanding of the family. The basis for creating the new subfamily is solely rooted in the cauda-like structure observed in male specimens. However, it is important to note that this feature is not unique to this subfamily as it has been partially recognized in certain genera of the family as mentioned in the Discussion section by the authors. In reality, the structure being referred to as the "unique cauda-liked structure" is actually the retractable aedeagus present in male specimens.
It is my recommendation that the author should gather additional information on these mites, particularly with regards to linking the males to their corresponding females. Based on this, they should reassess their taxonomic decision.
Author Response
Please see the attachment

Reviewer 3 Report (New Reviewer)
Dear Authors,
This is a very interesting article just so sorry it could not be complimented with females. The occurrence of similar structures in Dactyloscirus just makes me wonders about the uniqueness of this male structures. Please check it again. I will also send the manuscript with my corrections.

Author Response
Please see the attachment

Reviewer 4 Report (New Reviewer)
The manuscript " The Unique Cauda-liked Structure Represents a New Subfamily of Cunaxidae: Description of New Taxa and Discussion on Functional Morphology" is well prepared and worthy of publication in Animals.
It need only very small corrections.
I do not understand what means red colour in text – see in the manuscript.
Scale bars on some photos are nearly in the centre of photo – they should be placed near the border; some line of scale bars are too thick and some letters are too large.
Author Response
Please see the attachment

Reviewer 5 Report (New Reviewer)
Comments
The Unique Cauda-liked Structure Represents a New Subfamily of Cunaxidae: Description of New Taxa and Discussion on Functional Morphology
by
Jianxin Chen, Maoyuan Yao, Jianjun Guo, Tianci Yi and Daochao Jin
The Authors of the submitted paper present a cauda-like structure found in Cunaxidae. The new taxa: subfamily Cunaxicaudinae Chen & Jin subfam. nov., two new genera: Cunaxicaudus Chen & Jin gen. nov. (type genus) and Brevicaudus Chen & Jin gen. nov., and three new species: C. mohanensis sp. nov., C. neomohanensis sp. nov., and B. trapezoides sp. nov. are described. The Authors suggest and discuss, that the specialized cauda may be resulted by evolution of sperm transfer mode.
The paper is written in a standard array, materials and methods are described adequately, the figures and photographs are of proper quality. The reviewed paper represents a typical beta-taxonomic kind. Some corrections are marked in the enclosed text.
On the figure 1 (photograph), the end of xiphoid looks slightly different than on Figures 3 and 4 (drawings). Maybe it is the reviewer’s subjective feeling.
To conclude, I am of an opinion that the article fits into the scope of “Animals” and could be published, but after a minor corrections marked in the enclosed texts have been done.

Round 2
Reviewer 2 Report (New Reviewer)
It would be highly beneficial if the authors could gather information on the females, as creating a new subfamily based solely on male specimens cannot be justified.
This manuscript is a resubmission of an earlier submission. The following is a list of the peer review reports and author responses from that submission.
Round 1
Reviewer 1 Report
The taxonomic descriptions and illustrations in this paper are of high quality and the subject is extremely interesting and valuable. Unfortunately the paper has some very serious problems that make it unacceptable for publication. It is not acceptable for any mite taxon to be described from the males only. The authors should carefully search their Berlese funnel samples and identify all the female cunaxids, to find the females of these three species. The males described here may belong to species that are already known from the females.
The journal Animals is published in electronic form only, which does not comply with the International Code of Zoological Nomenclature. If this paper is published, it must be registered in ZooBank (web site zoobank.org)
The English language will need heavy revision. Most of the illustrations of Breucaudus are labelled Cunaxicaudus. The Latin word for short is brevis, not breuis.
The caudal appendage described here is probably the same as the appendage described in the male of other species, for example Dactyloscirus humuli. The caudal appendage in other species should be studied very carefully to determine the homology of different parts of the structure. If these structures are the same, there is no justification for describing a new subfamily and genus.
Reviewer 2 Report
The work is well designed and described.
The description of the specimens is sufficiently detailed, the drawings are excellent quality and comprehensive, the attached photos enhance the description of the collected specimens.
Small note regarding methodologies. If I did not miss it while reading, could the type of optics system used for the drawings and photographs be specified. Was a DIC, phase contrast, simple transmitted light used?
I do not consider the assessment on the English language in my domain, it seems acceptable.
Changes on the key (the original key modified by the authors): a reference to the drawings or photos in this paper could be added to the first dichotomous trait (line 498 of the pdf file), this would help to better understand the proposed taxonomic character.
the proposal for a new subfamily is supported by obvious morphological character, the epithets used for new species and subfamilies seem to me to meet the ICZN guidelines.
The finding of only males does not make the description of a new species conclusive, I agree with the authors' hypothesis: potential female specimens (not described) should differ little in appearance from male specimens, but it remains a hypothesis.
I am not a family Cunaxidae expert, I have personally tried to compare the descriptions of the specimens in this paper with descriptions of similar species, I found no evidence against the hypothesis of new species.
The work could be accepted, it would be ideal to find approval from other more qualified taxonomists to support this work.